# Drug repurposing to combat multidrug-resistant hookworm

Elise L. McKean[1], Catherine A. Jackson[2], Abigail Dowd[1], John M. Hawdon[2] and Damien M. O'Halloran[1],*

## ABSTRACT

Anthelmintic resistance (AR) poses an escalating threat to global soil-transmitted helminth control efforts, particularly in the context of mass drug administration (MDA) programs that rely heavily on benzimidazole drugs. To address the urgent need for novel therapeutics, a machine learning (ML) guided drug repurposing pipeline was developed to identify compounds effective against a multi-anthelmintic drug-resistant (MADR) isolate of *Ancylostoma caninum*. The computational strategy employed in this work involved training classifiers on curated datasets of anthelmintic actives and decoys, followed by screening all approved drugs from the online database, DrugBank. Selected compounds were tested using a tiered assay system comprising egg hatch assays (EHA), larval development assays (LDA), and adult survival assays. Among these, flutamide, a nonsteroidal antiandrogen approved for prostate cancer, demonstrated potent, egg- and larval-stage-specific activity against MADR hookworm. Flutamide exhibited a concentration-dependent inhibition of egg hatching and arrested larval development in MADR hookworm. These findings underscore the translational value of drug repurposing to accelerate therapeutic discovery against MADR helminths.

KEY WORDS: Hookworm, Drug repurposing, Anthelmintic resistance, *Ancylostoma caninum*, Machine learning

## INTRODUCTION

Hookworm infection is among the most widespread and debilitating neglected tropical diseases (NTDs), with nearly half a billion individuals estimated to be infected globally (Hotez et al., 2004; Bartsch et al., 2016). These parasitic nematodes are responsible for chronic blood loss, iron-deficiency anemia, fatigue, and impaired cognitive and physical development. The disease burden is heaviest in tropical and subtropical regions, particularly in impoverished communities lacking adequate sanitation infrastructure. Vulnerable populations, including children, pregnant women, and the elderly, bear the brunt of the disease's impact, suffering from long-term developmental and reproductive consequences (Bundy et al., 1995; Brooker et al., 2004; Pasricha et al., 2008). Current control efforts largely depend on mass drug administration (MDA) programs supported by the World Health Organization (WHO) and other global health stakeholders (Montresor et al., 2020). These programs rely on periodic distribution of benzimidazole anthelmintics, primarily albendazole and mebendazole, to at-risk populations regardless of individual infection status. The goal of MDA is to reduce morbidity rather than eliminate infection, and while this approach has significantly lowered the burden of disease in many regions, it does not interrupt transmission. Reinfection is common due to continued environmental contamination with infective larvae, inadequate water and sanitation infrastructure, and a lack of integrated control measures.

A growing concern in the parasitology and public health communities is the emergence and spread of anthelmintic resistance (AR) (Jimenez Castro et al., 2019). Although well-documented in veterinary medicine, where nematodes of livestock and companion animals such as *Haemonchus contortus*, *Trichostrongylus* spp., and canine hookworms have developed widespread resistance to all major classes of anthelmintics, evidence is now emerging that human parasitic nematodes may follow a similar trajectory (Kaplan, 2004; Coles et al., 2005; Gilleard, 2006; Kaplan and Vidyashankar, 2012; Kitchen et al., 2019; Orr et al., 2019; McKean et al., 2024). Genetic markers associated with benzimidazole resistance have been increasingly detected in human hookworm populations in endemic regions (Orr et al., 2019). These findings raise serious concerns about the long-term sustainability of current MDA strategies and the possibility of large-scale drug failure in the near future. Given the urgent need for new therapeutic options, there is growing interest in drug repurposing as a pragmatic strategy to identify anthelmintic candidates (Liu et al., 2019; Elfawal et al., 2019). Drug repurposing involves identifying new indications for existing drugs that have already been approved for other uses. This approach offers several advantages: the pharmacokinetics, safety, and manufacturing profiles of these compounds are already well-characterized, dramatically reducing the time and cost associated with bringing a new drug to clinical use. In the context of global health, where funding for NTD research is limited, repurposing represents an efficient and scalable solution. Recent advances in machine learning (ML) have further enhanced the feasibility of repurposing strategies (Chen et al., 2018; Vamathevan et al., 2019; King-Smith et al., 2024; Catacutan et al., 2024). ML algorithms can rapidly process large chemical libraries and biological datasets to predict drug-target interactions, rank compound efficacy, and prioritize candidates for *in vitro* and *in vivo* testing. Integrating cheminformatics with ML allows researchers to navigate the chemical space more effectively, uncovering subtle patterns and correlations that traditional screening might miss (Motaher et al., 2021; King-Smith et al., 2024).

In this study, a comprehensive ML-driven drug repurposing pipeline was applied to identify novel therapeutics for multi-anthelmintic drug-resistant (MADR) hookworm. A naturally occurring isolate of the canine hookworm *Ancylostoma caninum* (BCR), which exhibits resistance to benzimidazoles, tetrahydropyrimidines, and macrocyclic lactones – the three major anthelmintic drug classes used in both veterinary and human medicine – was utilized to test candidate compounds

[1]Department of Biological Sciences, The George Washington University, Washington, DC 20052, USA. [2]Department of Microbiology, Immunology and Tropical Medicine, The George Washington University, Washington, DC 20052, USA.

*Author for correspondence (damien@gwu.edu)

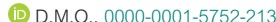 D.M.O., 0000-0001-5752-2131

Biology Open

(McKean et al., 2024). This isolate serves as an ideal model for investigating AR mechanisms and testing the efficacy of candidate compounds in a resistant background. The approach used herein involved training ML classifiers on curated datasets of known anthelmintic actives and structurally diverse decoys, which generated predictive models capable of screening repurposed candidates. Subsequent phenotypic screening of top-ranked compounds was conducted using a tiered assay system, including egg hatch assays (EHA), larval development assays (LDA), and adult survival assays across multiple hookworm species. From this approach, a nonsteroidal antiandrogen called flutamide, originally used to treat prostate cancer, emerged as a particularly promising compound. By combining computational prediction with experimental validation, we provide a scalable framework for identifying novel therapeutic leads with translational promise. These findings support further investigation of flutamide as an anthelmintic agent in naturally resistant isolates of hookworm, and advocate for expanded application of ML methodologies in parasitology research.

## RESULTS
### Computational prediction, filtering, and prioritization of repurposed candidates
We began our pipeline by training multiple classification algorithms on curated datasets comprising validated anthelmintic actives and decoys (Fig. 1). Molecular descriptors were used to generate a chemical feature matrix. Next, we downloaded all approved compounds available in DrugBank (Wishart et al., 2018; Knox et al., 2024) (accessed Aug 15, 2024), which we fed into classification algorithms to identify candidate anthelmintics from repurposed compounds. Consensus predictions were generated by retaining compounds that ranked as actives across the majority of

classification algorithms. To enhance biological relevance, we retrieved known drug targets for each compound and performed BLAST searches of these drug target sequences against the *A. caninum* genome (Camacho et al., 2009; Howe et al., 2017). Compounds whose targets shared significant homology (e-value≤0.0001) with *A. caninum* proteins were retained (Fig. 1). This refinement step yielded 1052 compounds that were then further filtered to include only approved, small molecules that exhibited zero Lipinski violations resulting in 478 compounds (Fig. 2). These compounds occupy diverse structural space (Fig. 2A,B). In Fig. 2A, compounds are ordered by hierarchical clustering of pairwise distances computed directly from atom-count vectors, and the color scale is explicitly defined such that dark blue indicates more similar compounds and dark red indicates more dissimilar compounds. Flutamide (PubChem CID 3397) is highlighted directly on the heatmap by a thickened black line allowing its distances to all other compounds to be readily visualized. In a three-dimensional atom-pair multidimensional scaling (MDS) representation derived from atom-pair fingerprints (Fig. 2B), flutamide's position is emphasized as a larger triangle with an arrow and text label to highlight its context within the final dataset of compounds we identified. Compounds that cluster together in this structural space tend to share related chemical features and, consequently, may share mechanisms of action or biological activity; some clusters are larger, suggesting common synthetic origins or widespread use of particular pharmacophores. A library of these compounds was ordered from MedChemExpress and delivered in 96-well format for screening.

### EHAs
Phenotypic screening began with EHA using eggs isolated from dogs infected with the BCR isolate of *A. caninum* (Fig. 3).

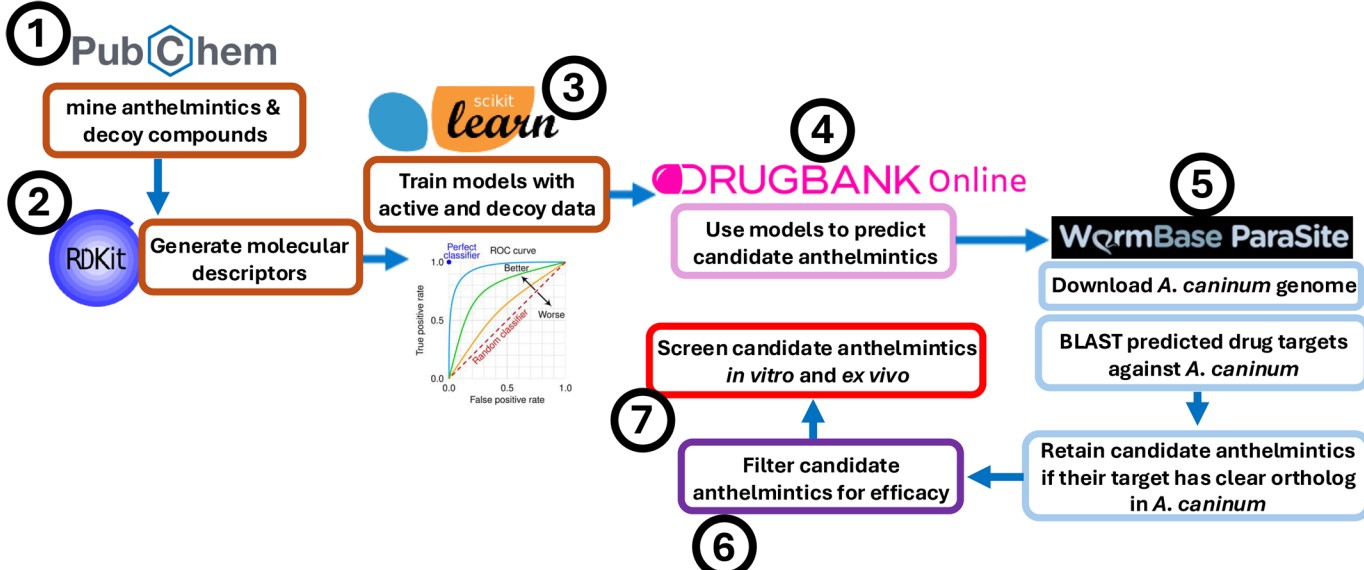

**Fig. 1. Cheminformatic and genomic pipeline used to prioritize repurposed anthelmintic candidates.** Schematic overview of the computational workflow used to identify candidate anthelmintics from approved drugs in DrugBank. Curated sets of putative anthelmintics (actives) and non-anthelmintics (inactives) were assembled from PubChem (Step 1) and encoded using RDKit-derived molecular descriptors (Step 2). Multiple machine-learning classifiers (support vector machines with linear and RBF kernels, k-nearest neighbors, Gaussian Naive Bayes, random forest, and decision tree) were trained and evaluated and model performance assessed (Step 3). A majority-vote ensemble was then applied to all approved DrugBank compounds (from *n*=2470 unique approved compounds with *n*=2966 annotated target proteins) to nominate predicted actives (Step 4). To increase biological plausibility and reduce false positives, the known targets of majority-vote-positive drugs (*n*=1151) were filtered by BLASTp homology against the *A. caninum* proteome, retaining only those with significant hookworm homologs (*n*=1052) (Step 5). The resulting high-confidence set of small-molecule candidates (*n*=478) was subsequently filtered for approved compounds with zero Lipinski violations (Step 6) and ordered from MedChemExpress for phenotypic screening in egg hatch, larval development, and adult survival assays (Step 7).

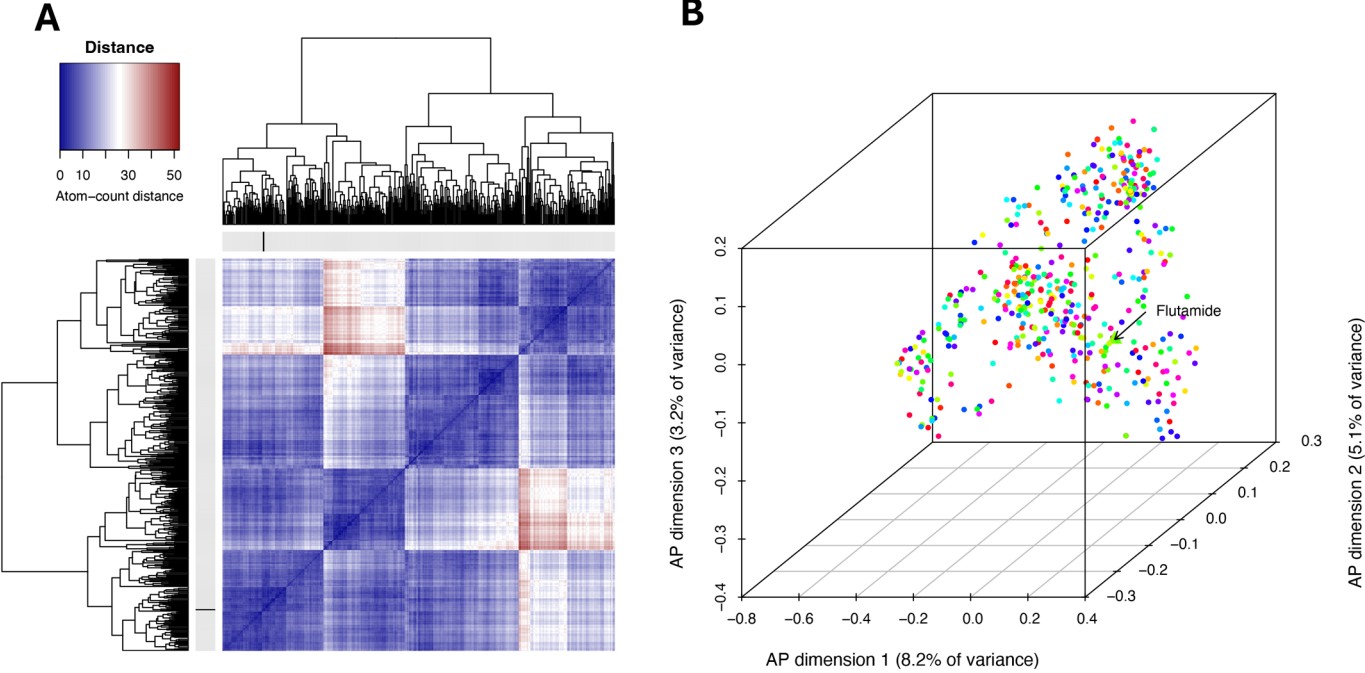

**Fig. 2. Visualization of the chemical space of the screened compounds with flutamide highlighted.** (A) Pairwise atom-count-based distance map for all compounds (*n*=478) in the SDF dataset. Distances were calculated as Euclidean distances between atom-count vectors (generated with atomcountMA in ChemmineR), and compounds were ordered by hierarchical clustering (average linkage). The heatmap color scale represents atom-count distance (dark blue=more similar; dark red=more dissimilar). The position of flutamide (PubChem CID 3397) is indicated directly on the matrix by thickened black line on the corresponding row and column. (B) An atom-pair distance matrix was computed from atom-pair fingerprints (ChemmineR sdf2ap), and classical MDS was used to embed all compounds into a three-dimensional atom-pair-based chemical space. The three axes represent the main dimensions of this space, with axis labels indicating the percentage of variance in the distance matrix explained by each dimension. Each point represents an individual compound and is colored according to its cluster assignment (cutoff=2). Flutamide is highlighted as a larger triangle and annotated with an arrow and text label to show its position within the overall compound space.

Flutamide emerged as a lead candidate, significantly inhibiting egg hatching in a concentration-dependent manner (Fig. 4). Flutamide is a non-steroidal anti-androgen whose primary mammalian target is the androgen receptor (UniProt ID P10275). The calculated $IC_{50}$ was 65.62 μM for drug resistant hookworm, with near-complete inhibition observed at 100 μM (Fig. 4, yellow line). No visible hatching or embryonation was detected at this dose (100 μM). Three biological replicates confirmed this result, and the compound demonstrated low variability between trials. Wild-type WMD *A. caninum* eggs were also included and yielded an $IC_{50}$ of 129.7 μM

(Fig. 4, blue line). Thiabendazole was also tested on BCR in this trial where we observed zero hatching at 12.5 μM, 25 μM, and 50 μM across three replicates that included 144, 142, and 146 eggs, respectively.

### LDAs

To determine whether flutamide affects later stages of development, LDAs were performed using the drug-resistant BCR isolate. Flutamide inhibited L1-L3 development with an $IC_{50}$ of 42.3 μM (Table 1). In controls, 84.8% of L1 developed to L3 after 7 days,

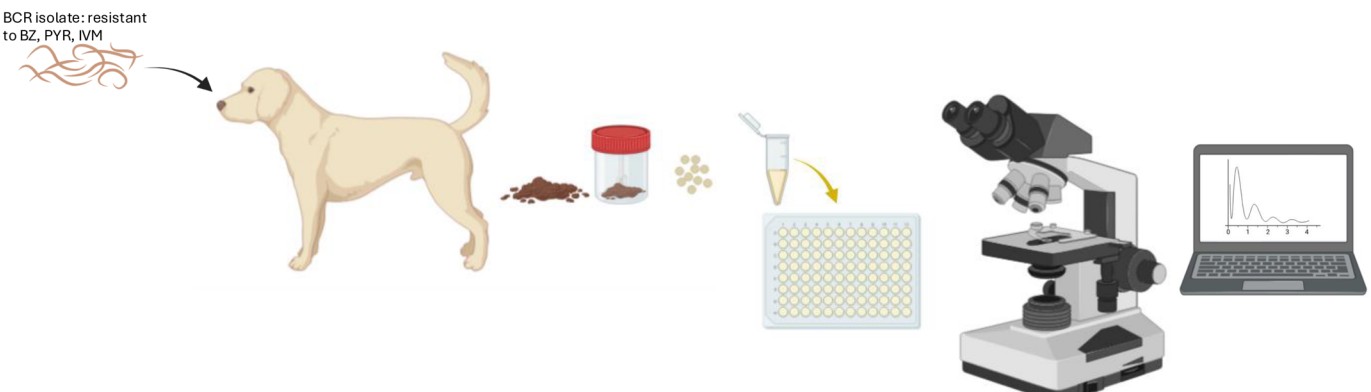

**Fig. 3. Overview of workflow for egg hatch and larval developmental assays.** Dogs were infected with multi-anthelmintic drug-resistant *A. caninum* (BCR), and the eggs were isolated from feces, washed, and incubated on 96-well plates for egg hatch phenotypic assays or cultured to the L1 stage for larval development assays. Egg hatch and larval development rates were ascertained microscopically. Within each independent experiment, each condition was run in triplicate wells (technical replicates) and repeated in three independent biological replicates. BZ, benzimidazole; PYR, pyrantel; IVM, ivermectin.

Biology Open

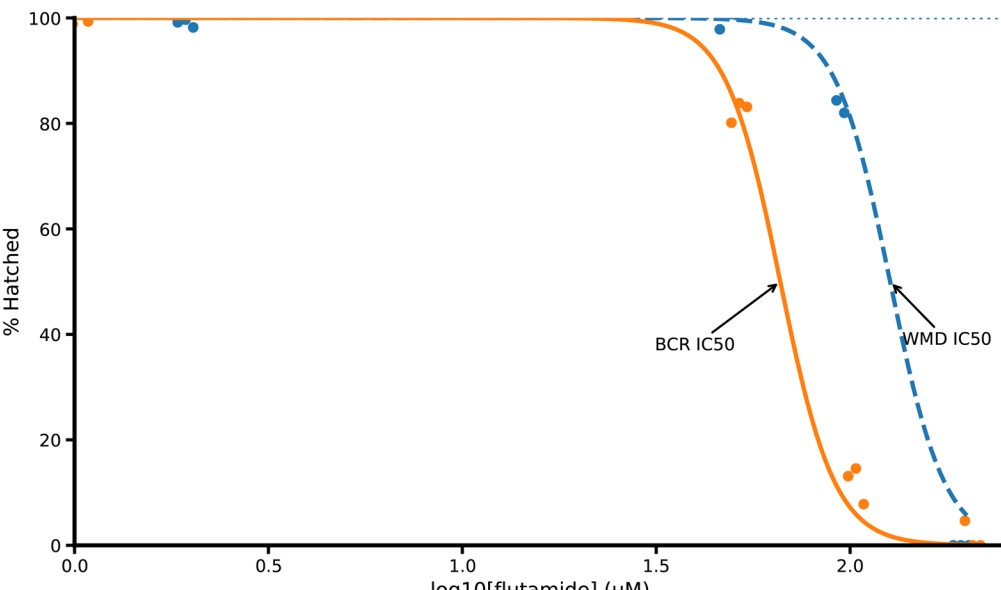

**Fig. 4. EHA for multi-anthelmintic drug-resistant hookworm (BCR, yellow color) and wild-type hookworm (WMD, blue color) revealed a potent inhibition in egg hatch.** Eggs were isolated from feces of infected dogs and dispensed into plates (approximately 50 eggs per well). Data represents three independent biological replicates per isolate, and within each biological replicate, each concentration was assayed in three technical replicate wells. Dotted line along the top indicates untreated negative control. Data are normalized to respective isolate controls. $IC_{50}$ BCR=65.62 µM and $IC_{50}$ WMD=129.7 µM.

with 9.2% at L2 and 6.0% at L1. Low flutamide concentrations (2-20 µM) produced a similar distribution (≥86% L3; ≤14% L1-L2). In contrast, at 50 µM no larvae reached L3, with 83.7% arrested at L2 and 16.3% at L1, and at 100 µM 19.4% remained at L2 and 80.6% at L1. Shapiro-Wilk tests indicated that percentage data for each stage deviated from normality (L1 $W=0.654$, $P=0.0029$; L2 $W=0.668$, $P=0.0044$; L3 $W=0.697$, $P=0.0089$), so Dunn's test was used for multiple comparisons. Higher flutamide concentrations (50 and 100 µM) significantly altered the distribution of larvae across stages compared with 2 and 20 µM ($P<0.05$ for all stages), demonstrating a potent inhibitory effect on development to L3 at 50-100 µM and to L2 at 100 µM.

### Adult hookworm survival assays

Next, flutamide was evaluated in *ex vivo* adult survival assays using the hookworm *Ancylostoma ceylanicum*. Adult-stage *ex vivo* assays were performed using *A. ceylanicum* rather than *A. caninum* as they are easily obtained from infected rodents and very closely related to *A. caninum*; using *A. ceylanicum* maintained in rodents allowed us to reduce dog use while still working with a closely related *Ancylostoma* species. Because adult worms are limiting, we focused on the concentrations that showed the strongest effects in the EHA and LDA (50 and 100 µM), alongside 1% DMSO and 10 nM emodepside controls, using one female and two males per condition. At both 24 and 48 h, all three worms in the DMSO control were alive. In the emodepside group, all three worms were dead at each time point. At 50 µM and 100 µM flutamide, all three

worms were alive at 24 h and at 48 h. Thus, emodepside reduced viability from 100% to 0% (3/3 worms dead) at both 24 and 48 h, whereas flutamide at 50 or 100 µM did not significantly reduce adult viability relative to the DMSO control (Fisher's exact tests: two-sided $P>0.05$; one-sided $P>0.05$).

### DISCUSSION

This study demonstrates the effectiveness of an ML-assisted drug repurposing pipeline in identifying novel anthelmintics active against MADR hookworm. The integration of computational screening with empirical validation allowed for the efficient prioritization and verification of candidates, culminating in the identification of flutamide as a promising lead compound. The successful identification of flutamide supports this computational approach and highlights the importance of combining *in silico* prediction with stage-specific phenotypic screening. In particular, the three-tiered assay design, incorporating EHAs, LDAs, and adult survival assessments, enabled us to evaluate the spectrum of efficacy and stage-specific action of candidate compounds. Flutamide consistently emerged as the top-performing compound during larval testing, with clear dose-dependent inhibition of hatching and development. The use of the BCR isolate of *A. caninum*, which exhibits natural resistance to three major anthelmintic classes, adds an important layer of translational relevance. This isolate emerged naturally and therefore serves as a valuable tool for testing next-generation compounds.

Current MDA programs rely almost exclusively on benzimidazoles such as albendazole and mebendazole, and their repeated use without rotation or combination therapy increases the likelihood of resistance emergence (Truscott et al., 2016; Coffeng et al., 2017; O'Halloran, 2021; Chong et al., 2021). Evidence of β-tubulin polymorphisms conferring resistance has already been reported in human hookworm populations (Orr et al., 2019). Our work suggests that flutamide may help circumvent existing resistance mechanisms in free-living stages of hookworm. While its mechanism of action in nematodes remains unknown, its consistent phenotypic effects suggest interference with developmental or neuromuscular pathways across egg-hatching and larval developmental stages. From our BLAST analysis using the annotated human androgen receptor protein sequence as a query against the predicted *A. caninum* proteome, we

**Table 1. Effects of flutamide concentration on the distribution of hookworm larval developmental stages (L1-L3)**

| Flutamide (µM) | L1% (n) | L2% (n) | L3% (n) |
|---|---|---|---|
| 0 (control) | 5.97% (13) | 9.19% (21) | 84.84% (192) |
| 2 | 6.09% (11) | 6.72% (12) | 87.19% (158) |
| 20 | 4.88% (10) | 8.20% (17) | 86.92% (179) |
| 50 | 16.26% (29) | 83.74% (150) | 0.00% (0) |
| 100 | 80.61% (126) | 19.39% (30) | 0.00% (0) |

Mean percentages were calculated from three separate replicate wells per flutamide concentration. Raw counts in parentheses (*n*) reflect the total number of larvae recovered across all replicates for each developmental stage. Larvae were scored after 7 days of culture.

identified a candidate nuclear hormone receptor in *A. caninum*, gene ANCCAN_07383, as the top hit. ANCCAN_07383 exhibits 67% identity at the protein level with NHR-69 in *C. elegans* and 93% protein identity to the *Haemonchus contortus* (ruminant parasite) ortholog, HCON_00017410. Although flutamide's target is well-characterized in mammals, there is no direct biochemical or genetic validation in parasitic nematodes confirming that this *A. caninum* nuclear hormone receptor is indeed the functional target, however, the ANCCAN_07383 ortholog in *C. elegans*, *nhr-69*, has been implicated in insulin signaling (Park et al., 2012). Insulin signaling has a well characterized role in growth and development in nematodes (Brand and Hawdon, 2004; Murphy and Hu, 2013). Taken together, we propose a model whereby flutamide could inhibit a regulator of insulin signaling in *A. caninum*, which perturbs hatching and development as observed in our data. However, further testing would be required to determine this mechanism. Moreover, testing of flutamide against the adult hookworm stage would be required to search for other concentrations or phenotypic effects that could prove useful, however, flutamide's pharmacokinetics, absorption, distribution, metabolism, and excretion (ADME) profiles in mammalian hosts would also need to be assessed in the context of enteric nematode infection. While flutamide is well-tolerated in humans when administered systemically, its efficacy against gut-dwelling parasites will depend on its bioavailability and intestinal retention. Formulation improvements or delivery strategies may be necessary to optimize its anthelmintic activity in addition to further testing of the adult stages. After standard oral doses in humans, flutamide and its active metabolite 2-hydroxyflutamide reach peak plasma concentrations in the low micromolar range ($\approx$0.1 µM and 4-8 µM, respectively) (Schulz et al., 1988), which is well below the $IC_{50}$ values observed in our assays. While intestinal luminal concentrations may transiently be higher than plasma, we do not regard 100 µM as a clinically realistic exposure. Our data therefore indicate that flutamide serves primarily as a tool compound that reveals a hormonally regulated developmental pathway in hookworms, rather than as a direct candidate for therapeutic repurposing.

In conclusion, these findings demonstrate the utility of combining ML with phenotypic screening to identify repurposed anthelmintics for combating resistance in parasitic nematodes. As AR continues to jeopardize the efficacy of existing anthelmintics and control programs, such integrative approaches will be critical to sustaining progress in helminth control and elimination.

## MATERIALS AND METHODS
### Cheminformatic pipeline
List of putative anthelmintics (actives) and non-anthelmintics (decoys or inactives) were obtained by searching PubChem (Kim et al., 2021) for compounds tagged with the terms 'anthelmintic' and/or 'anti-parasitic' (for the active list), or for compounds that did not contain these terms for the non-active list (Table S1). We wanted the lists to be of comparable sizes, and that the compounds represented similar Tanimoto distributions (less than 0.2 for most compounds; Fig. S1). Molecular descriptors were calculated using RDKit (Lovrić et al., 2019) to generate numerical representations of each molecule (Tables S2-S3). These datasets of descriptors were combined to train classification models capable of predicting novel candidate repurposed anthelmintics. Multiple classification algorithms were implemented using the scikit-learn library in Python (Pedregosa et al., 2011). These included: Linear Support Vector Classifier (SVC, linear kernel); SVC with RBF kernel; k-Nearest Neighbors (KNN, k=5, Minkowski metric, $P$=2); Gaussian Naive Bayes; Random Forest ($n$_estimators=10, criterion="entropy", max_features="auto"); and Decision Tree (criterion="entropy"). An 80/20 train-test split was used (train_test_split, test_size=0.2, random_state=0) and

applied fivefold cross-validation to estimate model performance. Model performance was examined based on sensitivity (recall), specificity, precision, accuracy, F1 scores, AUC (Figs S2-S3), and the confusion matrix for each classifier (Fig. S4). A majority vote across the models was used to prioritize approved compounds from the drug database DrugBank (Wishart et al., 2018; Knox et al., 2024). This ensemble approach reduced false positives and improved overall prediction reliability. To improve the specificity of predictions and reduce off-target effects, predicted drug targets were compared against the *A. caninum* genome (PRJNA72585 WBPS19) (Howe et al., 2017). Protein sequences of known drug targets from all the available approved DrugBank compounds were downloaded from DrugBank, and then filtered to only include those that passed the majority vote from the classification algorithms; the target proteins from this filtered list were subjected to BLASTP (Camacho et al., 2009) searches against the *A. caninum* genome. This equated to 2470 unique compounds and 2966 unique targets, and after filtering for compounds that passed the majority vote became 1151 unique compounds. BLAST was performed using local blastp (NCBI BLAST+ 2.10.0+) executable with the following parameters: default scoring matrix: BLOSUM62; E-value cutoff: 1e-4; low-complexity filtering: SEG enabled (default); gap penalties: defaults for BLOSUM62. Matches with e-values >0.0001 were discarded, retaining only those drug candidates whose targets had significant sequence similarity to hookworm homologs. This lowered the filtered list of compounds from 1151 to 1052. Next, we mapped these 1052 DrugBank compounds to ChEMBL and retained only approved, small compounds that exhibited zero Lipinski violations, which then yielded 478 compounds (Table S4). This dual-layer screening process generated a refined list of high-confidence candidates for phenotypic screening against MADR hookworm.

### Parasites
The wild-type laboratory isolate of *A. caninum* named WMD (US National Parasite Collection No. 106970) was isolated from a rescued labrador retriever in 2013 and maintained by passage 2-4 times per year. An MADR isolate of *A. caninum* named BCR was isolated from a retired racing greyhound in October 2019 (McKean et al., 2024). Infective third-stage larvae (iL3s) reared from eggs collected from feces of the infected dog were used to infect a hookworm naïve beagle. All isolates were maintained in beagles as described previously (Schad and Page, 1982; Krepp et al., 2011). An Indian strain of *A. ceylanicum* (US National Parasite Collection No. 102954) was maintained in immunocompromised mice and Syrian golden hamsters as described previously (Langeland et al., 2023; 2024).

### Reagents
Thiabendazole, emodepside, and dimethyl sulfoxide (DMSO) were purchased from Sigma-Aldrich (St. Louis, MO, USA). A curated selection of chemical reagents was obtained from MedChemExpress (Monmouth Junction, NJ, USA) to facilitate the screening of small molecules for anthelmintic activity against hookworm. Compounds that were either not available in solution or classified as DEA-controlled were excluded. Each compound was solubilized in DMSO and diluted to appropriate working concentrations for use in 96-well plates, allowing for simultaneous testing of multiple compounds under standardized conditions. Flutamide was also purchased separately for testing from MedChemExpress.

### EHA
To assess compound efficacy against the earliest stage of hookworm development, EHA was conducted using eggs isolated from feces of dogs infected with the MADR *A. caninum* BCR isolate and wild-type WMD isolate as described previously (Kitchen et al., 2019; McKean et al., 2024). Fecal material was homogenized in water and filtered through successive sieves to remove debris. The egg suspension was subjected to salt flotation and three washes in 15 ml of BU (50 mM $Na_2HPO_4$, 22 mM $KH_2PO_4$, 70 mM NaCl, pH 6.8) before being counted (Hawdon and Schad, 1991). Eggs were dispensed at approximately 50 per well into 96-well flat-bottom plates. Test compounds were added to the wells along with positive (thiabendazole) and negative (vehicle) controls. Plates were incubated at 27°C for 48 h. After incubation, hatching was assessed microscopically, and hatch rates were determined by counting larvae versus unhatched eggs. Half

maximal inhibitory concentration ($IC_{50}$) values were calculated using non-linear regression in GraphPad Prism (ver. 10, GraphPad Software San Diego, CA, USA).

## LDA

LDA was performed to evaluate effects of compounds on post-embryonic development. Eggs were allowed to hatch in BU supplemented with 100 U/ml Pen-strep, 0.1 mg/ml kanamycin, and 0.1 mg/ml ampicillin for 24 h at 27°C before being transferred to assay plates (McKean et al., 2024). L1 larvae were cultured at 27°C for 7 days in BU supplemented with test compounds and $1.56 \times 10^8$ cells of *Escherichia coli* OP50 bacteria as a food source. After incubation, Lugol's iodine was added to each well to fix larvae and facilitate counting. The numbers of L1, L2, and L3 stages were counted microscopically. The percentage of developed L3 was calculated using the formula:

$$\frac{L3}{L1 + L2 + L3} \times 100,$$

where $L_X$ denotes the total number of larvae at stage $x$. Dose-response curves were plotted, and $IC_{50}$ values were determined for each compound. The average percentage of larvae developing to the L3 stage was plotted against the log of the drug concentration, and the curve analyzed with the sigmoidal dose-response (variable slope) function in GraphPad Prism to generate the $IC_{50}$ and the Hill slope value.

## Adult worm survival assay

To evaluate compound efficacy against adult-stage hookworms, we performed *ex vivo* survival assays on *A. ceylanicum* adults as previously described (Langeland et al., 2023). Infected animals were euthanized and the intestines excised, opened longitudinally, and incubated in warm PBS to allow the worms to release from the intestine. Worms were hand-picked into microfuge tubes, washed eight times in 1.5 ml of RPMI, and transferred to 24-well plates containing RPMI-1640 supplemented with antibiotics (80 U/ml penicillin, 80 µg/ml streptomycin, 80 µg/ml gentamicin), an antifungal (5 µg/ml amphotericin B), and test compounds. Worms of each sex were incubated at 39°C. Controls included media alone or the anthelmintic emodepside (Keiser and Häberli, 2021). Viability was monitored at 24 h and 48 h using motility scoring following probing with a fine needle. Worms that did not respond to stimulation were scored as dead.

## Statistical analysis

*In vitro* EHA and LDAs were conducted in triplicate with three independent biological replicates and *ex vivo* adult assays included a minimum of three adults per treatment and at least one individual from each sex. Sample sizes were chosen *a priori* based on established assay practice, feasibility, and availability of parasite material, and the exact sample size for each analysis is reported in the corresponding figure legend. For summary plots, technical replicates were averaged within each independent experiment, and error bars (when shown) reflect variation across independent biological replicates. Where feasible, plots display individual biological replicate values to show the data spread. Inclusion/exclusion criteria were pre-established: data were excluded only when wells or samples were compromised (e.g. contamination, evaporation, or handling errors that prevented reliable scoring); otherwise, all collected data were included. Treatments were assigned to wells using a pre-specified plate map with interleaved vehicle controls to minimize positional effects; no formal randomization was used. Investigators were not blinded to treatment during scoring; outcomes were based on objective counts and staging criteria applied uniformly across conditions. Data were analyzed using GraphPad Prism. $IC_{50}$ values were estimated using nonlinear regression with a sigmoidal dose-response model. Data were tested for normality using the Shapiro-Wilk Test followed by one-way ANOVA and Tukey's post hoc test or Kruskal–Wallis with Dunn's correction for nonparametric data. Significance was set at $P<0.05$. ML model performance metrics and enrichment scores were calculated using scikit-learn (v1.2.2) in Python following best practices (Wilson et al., 2014).

## Ethical Standards

All experiments involving animals were conducted in strict accordance with the recommendations of the National Institutes of Health (USA) Guide for the Care and Use of Laboratory Animals and the Animal Welfare Act (National Research Council, 2011). The animal protocol used (A2024-107) in this study was approved by the Institutional Animal Care and Use Committee of The George Washington University (USA).

### Acknowledgements
The authors would like to acknowledge the computing resources provided on the High Performance Computing Cluster operated by Research Technology Services at the George Washington University, and the animal husbandry and veterinary service support provided by the Office of Animal Research at the George Washington University.

### Competing interests
The authors declare no competing or financial interests.

### Author contributions
Conceptualization: E.L.M., J.M.H., D.M.O.; Data curation: E.L.M., C.A.J., D.M.O.; Formal analysis: E.L.M., C.A.J., D.M.O.; Funding acquisition: J.M.H., D.M.O.; Investigation: E.L.M., C.A.J., A.D., D.M.O.; Methodology: E.L.M., C.A.J., D.M.O.; Project administration: J.M.H., D.M.O.; Resources: D.M.O.; Software: D.M.O.; Supervision: J.M.H., D.M.O.; Validation: A.D., D.M.O.; Visualization: D.M.O.; Writing – original draft: E.L.M., D.M.O.; Writing – review & editing: E.L.M., C.A.J., A.D., J.M.H., D.M.O.

### Funding
This research was supported by the Wilbur V. Harlan Fund to the Department of Biological Sciences at The George Washington University and grant R21AI176230 to J.M.H. from the National Institutes of Allergy and Infectious Diseases of the National Institutes of Health. The sponsors had no role in the study design, collection, analysis or interpretation of data, writing the manuscript or the decision to submit the manuscript for publication. Open Access funding provided by George Washington University. Deposited in PMC for immediate release.

### Data and resource availability
All scripts, configuration files, and input data necessary to reproduce the cheminformatics, machine-learning, and BLAST analyses described in this manuscript are available at the following GitHub repositories: https://github.com/dohalloran/BiologyOpen-ML and https://github.com/dohalloran/BiologyOpen-BLAST. All relevant data and details of resources can be found within the article and its supplementary information.

### Peer review history
The peer review history is available online at https://journals.biologists.com/bio/lookup/doi/10.1242/bio.062380.reviewer-comments.pdf

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
