## [Peer Review File · Biology Open]

Drug repurposing to combat multidrug-resistant hookworm

Elise L. McKean, Catherine A. Jackson, Abigail Dowd, John M. Hawdon and Damien O'Halloran

DOI: 10.1242/bio.062380

Editor: Christopher A. Maher

Review timeline

Original submission: 21 November 2025

Editorial decision: 1 December 2025

First revision received: 9 January 2026

Accepted: 16 January 2026

Original submission

First decision letter

MS ID#: bio.062380

MS Title: Drug repurposing to combat multidrug-resistant hookworm

Authors: Elise L. McKean, Catherine A. Jackson, Abigail Dowd, John M. Hawdon and Damien O'Halloran

I have now reached a decision on the above manuscript.

The reviewer reports are shown at the bottom of this email.

As you will see, the reviewers raised a number of substantial criticisms that prevent me from accepting the paper at this stage.

They suggest, however, that a revised version might prove acceptable, if you can address their concerns. If you think that you can deal satisfactorily with the criticisms on revision, I would be pleased to see a revised manuscript. We would then return it to the reviewers.

At this stage, we also ask you to ensure your manuscript complies with our formatting guidelines. Provided you are able to fully address the referees' comments, we are positive about publication of your paper (we accept over 95% of revision submissions) and therefore hope you won't mind any extra work involved in reformatting your manuscript at this point.

Please upload both a 'clean' version of your Word file, along with a highlighted version clearly showing where you have made changes in the revised manuscript. Please avoid using 'Track changes' in Word files as these are lost in PDF conversion.

I should be grateful if you would also provide a point-by-point response detailing how you have dealt with the points raised by the reviewers in the 'Response to Reviewers' box. Please attend to all of the reviewers' comments. If you do not agree with any of their criticisms or suggestions please explain clearly why this is so.

Reviewer 1

Comments for the author

© 2026. Published by The Company of Biologists under the terms of the Creative Commons Attribution License (<https://creativecommons.org/licenses/by/4.0/>).

The manuscript by Elise McKean et al describes a multi-disciplinary approach to drug repurposing that combines training of machine-learning models, bioinformatics, cheminformatics, and experimental assays to validate hit compounds. In a well-written and focused paper they show that flutamide (anonsteroidal antiandrogen) has an anthelmintic effect, as shown in diverse assays. And they also show flutamide is also active against drug-resistant worms.

MAJOR ISSUES

As mentioned, the manuscript lacks some essential details that would aid others to reproduce the pipeline in their own settings, either using the same data or their own data for other organisms or initial sets of actives + decoys.

Lack of data (input). No data is provided regarding the initial set of curated anthelmintic compounds. This is one of the main deliverables of the paper!

Lack of details in methods. How are decoys selected? Were they generated instead from actives? How?

Please provide the initial set of actives + decoys as supporting data.

lines 147-150, "Molecular descriptors and chemical fingerprints were calculated using PaDEL-Descriptor (Yap, 2011) and RDKit (Lovrić et al., 2019)" which feature was calculated with which software package? For reproducibility it would be nice perhaps to list each feature in a table, and use the columns to specify which package, command, specific parameters or arguments were used in each case.

lines 153-154, "multiple classification algorithms" Which ones? Please provide a list of the algorithms tested as well as the used parameters and hyperparameters (if applicable) in each case.

lines 154-155, how many folds were used in k-fold cross-validation? what was the ratio of train/test splits?

lines 161-166, and also lines 187-189. What is the source of "protein sequences of known drug targets"? How many were them? What is the diversity of protein classes in this dataset?

Nor the main text, nor Fig 1 clarify what is the source of the candidate drug targets. Was it DrugBank?

Please provide parameters for the BLAST search against *A. caninum* genome. Which scoring matrix was used? BLOSUM45? 62? 80? Without knowing this, the E-value threshold (0.0001) is difficult to ascertain. Other important parameters?

Lack of data (output, results). The machine learning pipeline is not clearly defined (no classification algorithms are explicitly mentioned, hence affecting reproducibility by others, and no description or list of parameters used and their values is provided. Further, as it stands the manuscript is not providing any support (either as tables, figures or supplementary materials) showing the performance assessment of each algorithm or the consensus model.

Please provide an annotated notebook or code to allow others to re-run the pipeline. Either a jupyter or quarto notebook with code + text + figures, or just plain code in a publicly accessible repository (github, gitlab) should suffice.

Please provide plots of ROC curves, precision-recall curves, plots of F1 scores, or tables showing global sensitivity, specificity, confusion matrices, AUC values, or better AUC focused on the initial part of the ROC curve, to show the performance at the task of finding hits from the top of the ranking (e.g. pAUC 0.1-0.2, see partial Area Under the ROC Curve, McClish, DK (1989) "Analyzing a portion of the ROC curve". Medical Decision Making 9: 190-195.)

As for any machine learning paper, these plots and metrics are what the readership expects to validate the findings and pipeline.

The same applies to the consensus model. Why define consensus as 'those compounds classified as actives by at least two models'? Why not treat the ensemble/consensus stage as another machine learning process and measure performance of different consensus building strategies (at least 3 models? Use threshold in each algorithm as a score and also find the best threshold in each case, then select the best-performing one? If the best-performing is the one reported, then show the supporting metrics, plots, etc to validate this way of consensus building.

Lack of a simple baseline for comparison. The proposed pipeline is based on the hypothesis that known anthelmintic compounds have chemical features or properties that can be learned by machine learning algorithms. While this is certainly true in most cases, it is also true that simple chemical similarity searches also work to retrieve good candidates (analogs) from chemical databases. The performance of machine-learning vs simple chemical similarity can be tested and compared. I suggest the authors to perform some basic chemical similarity searches between their actives and decoys against DrugBank using a selection of different fingerprints and maybe also similarity metrics (see Safizadeh et al 2021, <https://pubs.acs.org/doi/10.1021/acs.jcim.0c00993>), and then compare the confusion matrices obtained against the one obtained from their ensemble method. How many TPs (true positives), TNs, FPs, FNs are there in each case? Providing a simple baseline for comparison is a nice way to show the improved performance of machine learning methods, and is also an elegant way to present and showcase the need for more complex methods to readers.

How does flutamide rank in each case?

BLAST searches against the *A. caninum* genome. I agree that both having an active compounds + putative mechanism of actions (targets) is beneficial for prioritizing or filtering further a library of targets, the original targets in DrugBank may come from organisms that may be evolutionary distant in relation to *A. caninum*. So, regarding the BLAST searches: you may also be interested in trying to capture more distant hits by using profile-HMM searches (e.g. using profile-HMMs derived from Pfam domains in the known targets as queries) to search against proteins encoded in the *A. caninum* genome. This may or may not reveal additional conserved targets but it may be worth trying. This is just a suggestion.

Also related to the targets of compounds: you use this as a filter strategy but then do not mention anything about the targets (e.g. which is the target of flutamide? what is the retrieved hit from the *A. caninum* genome? why do you say that "While its mechanism of action in nematodes remains unknown, its consistent phenotypic effects suggest interference with developmental or neuromuscular pathways across egg-hatching and larval developmental stages." (lines 288-291). What are the hypothesized activities for the *A. caninum* candidate target? Is it conserved in other helminths? (nematodes, flatworms)?

Lipinski violations: I understand the need to further refine and reduce the number of prioritized compounds to a manageable number. We've all been there. That said, it would be a nice addition to the paper to analyze and show the features and properties of known anthelmintics in relation to Lipinski rules before applying filters. See for example the analysis of anti-tuberculosis drugs here: Motamen S, Quinn RJ. Analysis of Approaches to Anti-tuberculosis Compounds. ACS Omega. 2020 Oct 27;5(44):28529-28540. doi: 10.1021/acsomega.0c03177. Perhaps there are some violations to Lipinski's rules that may be admitted or less penalized in the case of anthelmintics? Also a suggestion here.

Reviewer 2

Comments for the author

This manuscript, titled "Drug Repurposing to Combat Multidrug-Resistant Hookworm," addresses a relevant and timely issue due to the growing concern about anthelmintic resistance. However,

several methodological, interpretative, and presentation issues need to be addressed before the manuscript is suitable for publication.

Major corrections

1. Please provide sufficient details on the machine learning methods to enhance clarity and reproducibility of the study. Missing information includes the number of active vs. decoy sets used, the feature selection or preprocessing pipeline, model performance metrics for each model, and validation on external test sets. If possible, provide the full list of top predicted compounds as supplementary materials and the reasons for selecting only the ones tested. Additionally, the cheminformatics methods should precede the three-tiered phenotypic assay, given that compounds derived from ML were used in the assay. This arrangement will also align with the results presented.

2. It is worrisome for the authors to report that "WMD was isolated from a rescued Labrador retriever in 2013 and maintained by continuous passage, while BCR was isolated in October 2019." If WMD was maintained by continuous passage from 2013, what is the passage number in 2024 when the study was conducted? Passaging an organism continuously for over 10 years may not follow best practice, which calls for using an organism with a low passage number.

3. To add experimental rigour and ensure the reproducibility of the assays, specify the volume of BU and the different concentrations of each test compound used for the egg hatch assay. In the adult worm survival assay, the authors reported that "Worms were then washed and transferred to 24-well plates containing RPMI-1640 supplemented with antibiotics and an antifungal." Please specify what was used in the wash, the volume of RPMI added to each plate, and the names of the antibiotics and antifungal used to supplement RPMI. Furthermore, state the rationale for using *A. ceylanicum* adults only instead of *Ancylostoma caninum* in the adult worm survival assay.

4. There is a big gap between the protocols reported in the methods and the results presented. The results of all the assays focused on flutamide, while the other test compounds and standard anthelmintics included in the assay protocols were not provided. Please provide the results of the positive-control EHA and LDA curves for standard anthelmintics for comparison. Then, compare the efficacy of standard anthelmintics used across each assay with that of flutamide. Furthermore, provide supplementary tables containing a full list of all filtered DrugBank compounds, their ML prediction scores, and the raw phenotypic assay counts for all compounds tested. Also, avoid repetition and reduce redundancy by summarizing the results of the larval development assays since the table already contains values repeated in the interpretation.

5. Importantly, although the figures were mentioned in the results interpretation, the figure labels were not placed side by side with the figure presented. Improve the readability and interpretation of Figures 1 and 2 by adding legends, clearer colour scales, axis labels, cluster names, or representative structures. In Figure 4, label the IC₅₀ values of flutamide in WMD and BCR assays visually. The results of the adult worm survival assay were not presented in any table or figure. In the discussion section, please include possible reasons for the activity observed with flutamide, considering its known activities and metabolites, as well as the relevance of anti-androgenic mechanisms of action to nematodes. It is highly recommended that the authors compare the IC₅₀ values obtained with flutamide with those of known anthelmintics across all the assays. This aspect is imperative because the IC₅₀ values for flutamide appear significantly higher than those of typical anthelmintics. Moreover, the treatment concentrations used in the assay (up to 100 µM) are high and may lack physiological relevance. Explain if these levels are achievable in vivo. Furthermore, *Sinceno* experiments on *A. caninum* adults (the focal species) was done. The authors should explain if the results of the *A. ceylanicum* assay could be extrapolated to *A. caninum* or they should acknowledge this as a limitation of the findings. Finally, propose specific future research directions needed to confirm the anthelmintic effects of flutamide.

Minor corrections

- i. Italicize all the *in vivo* and *in vitro* where they appeared in the study.
- ii. Add the ethical approval number in the ethical standards section.
- iii. Lines 105 and 106: Justify why compounds not available in solution were excluded, since it was stated that "DMSO was used to solubilize the compounds".
- v. Use Microsoft Equation Editor to represent the formula for calculating the percentage of developed larva $(L3_{total} / (L1 + L2 + L3) \times 100)$.

Reviewer's Responses to Questions

Experimental quality

Does each figure have the proper controls?

If 'No', please indicate reasons in Comments for Author box below.

Reviewer #1:

- Yes

Reviewer #2:

- No
-

Were the data analyzed using appropriate statistical tests?

If 'No', please indicate reasons in Comments for Author box below.

Reviewer #1:

- Yes

Reviewer #2:

- Yes
-

Reproducibility

Were experiments performed using adequate number of biological replicates?

If 'No', please indicate reasons in Comments for Author box below.

Reviewer #1:

- Yes

Reviewer #2:

- Yes
-

Does the methods section provide sufficient detail to permit reproducibility?

If 'No', please indicate reasons in Comments for Author box below.

Reviewer #1:

- No

Reviewer #2:

- No
-

Completeness

Are the manuscript's conclusions supported by the data?

If 'No', please indicate reasons in Comments for Author box below.

Reviewer #1:

- Yes

Reviewer #2:

- Yes
-

Scholarship

Do the authors cite and discuss the merits of data that would argue for and against their conclusion?

If 'No', please indicate reasons in Comments for Author box below.

Reviewer #1:

- Yes

Reviewer #2:

- Yes

Does the manuscript title & abstract accurately reflect the contents of the manuscript, without hyperbole?

If 'No', please indicate reasons in Comments for Author box below.

Reviewer #1:

- Yes

Reviewer #2:

- Yes
-

First revision

Author response to reviewers' comments

We thank the Reviewers for their thoughtful and constructive comments on our manuscript “Drug Repurposing to Combat Multidrug-Resistant Hookworm.” We have revised the manuscript substantially to address all points raised, and we have also made our full analysis pipelines and data publicly available in two GitHub repositories:

- Machine learning & cheminformatics pipeline:
<https://github.com/dohalloran/BiologyOpen-ML>
- BLAST-based target mapping and downstream analyses:
<https://github.com/dohalloran/BiologyOpen-BLAST>

These repositories contain all scripts, configuration files, and input data necessary to reproduce the cheminformatics, machine-learning, and BLAST analyses described in the manuscript.

Below we respond to each reviewer point-by-point.

REVIEWER 1

Comment 1. *Lack of data (input). No data is provided regarding the initial set of curated anthelmintic compounds. This is one of the main deliverables of the paper!*
Lack of details in methods. How are decoys selected? Were they generated instead from actives? How?

Please provide the initial set of actives + decoys as supporting data.

Response 1: We agree that these details and datasets are central deliverables. We now provide three new supplemental tables: **Table S1:** includes a complete list of the active and inactive compounds that were used; **Table S2:** contains a complete list of all the descriptors used for the active dataset; **Table S3:** contains a complete list of all the descriptors used for the inactive dataset. Furthermore, as mentioned above we have developed two GitHub repos that contain the complete set of data, scripts, and configurations used to generate these data (see section 4 entitled “4. Generating RDKit descriptors” of the following repo:

<https://github.com/dohalloran/BiologyOpen-ML>). In the revised manuscript we also explain that the inactive molecules were selected to have similar Tanimoto distribution to that of the active dataset and provide a new supplemental figure (**Figure S1**) showing these distributions.

We have now explicitly described the above details on lines 221- 239 as follows:

“List of putative anthelmintics (actives) and non-anthelmintics (decoys or inactives) were obtained by searching PubChem (Kim *et al.*, 2021) for compounds tagged with the terms “anthelmintic” and/or “anti-parasitic” (for the active list), or for compounds that did not contain these terms for the non-active list (Table S1). We wanted the lists to be of comparable sizes, and that the compounds represented similar Tanimoto distributions (less than 0.2 for most compounds: Fig. S1). Molecular descriptors were calculated using RDKit (Lovrić *et al.*, 2019) to generate numerical representations of each molecule (Table S2-S3). These datasets of

descriptors were combined to train classification models capable of predicting novel candidate repurposed anthelmintics. Multiple classification algorithms were implemented using the scikit-learn library in Python (Pedregosa *et al.*, 2011). These included: Linear Support Vector Classifier (SVC, linear kernel); SVC with RBF kernel; k-Nearest Neighbors (KNN, k = 5, Minkowski metric, p = 2); Gaussian Naive Bayes; Random Forest (n_estimators = 10, criterion = "entropy", max_features = "auto"); and Decision Tree (criterion = "entropy"). An 80/20 train-test split was used (train_test_split, test_size = 0.2, random_state = 0) and applied 5-fold cross-validation to estimate model performance. Model performance was examined based on sensitivity (recall), specificity, precision, accuracy, F1 scores, AUC (Figs S2-S3), and the confusion matrix for each classifier (Fig. S4). A majority vote across the models was used to prioritize approved compounds from the drug database DrugBank (Wishart *et al.*, 2018; Knox *et al.*, 2024)".

Comment 2. lines 147-150, "Molecular descriptors and chemical fingerprints were calculated using PaDEL-Descriptor (Yap, 2011) and RDKit (Lovrić *et al.*, 2019)" which feature was calculated with which software package? For reproducibility it would be nice perhaps to list each feature in a table, and use the columns to specify which package, command, specific parameters or arguments were used in each case.

Response 2: Please refer to our response to comment 1 above in relation to Tables S1-S3 and the GitHub repo section 4 entitled "4. Generating RDKit descriptors": <https://github.com/dohalloran/BiologyOpen-ML>

Comment 3. lines 153-154, "multiple classification algorithms" Which ones? Please provide a list of the algorithms tested as well as the used parameters and hyperparameters (if applicable) in each case.

Response 3: We have expanded the Methods on lines 230-237 to explicitly list all models and their parameters which included the following classification algorithms:

Linear **Support Vector Classifier** (SVC, linear kernel)

SVC with RBF kernel

k-Nearest Neighbors (KNN, k = 5, Minkowski metric, p = 2)

Gaussian Naive Bayes

Random Forest (n_estimators = 10, criterion = "entropy", max_features = "auto")

Decision Tree (criterion = "entropy")

All models were implemented using **scikit-learn**, and the training code including model definitions, parameters, preprocessing, and evaluation is available in the **BiologyOpen-ML** repository (<https://github.com/dohalloran/BiologyOpen-ML> à train_activity_models.py and associated helper modules).

Comment 4. lines 154-155, how many folds were used in k-fold cross-validation? what was the ratio of train/test splits?

Response 4: We now state explicitly in the Methods section under "Cheminformatic pipeline" on lines 233- 235, that we used "An 80/20 train-test split was used (train_test_split, test_size = 0.2, random_state = 0) and applied 5-fold cross-validation to estimate model performance".

Comment 5. lines 161-166, and also lines 187-189. What is the source of "protein sequences of known drug targets"? How many were them? What is the diversity of protein classes in this dataset?

Nor the main text, nor Fig 1 clarify what is the source of the candidate drug targets. Was it DrugBank?

Please provide parameters for the BLAST search against *A. caninum* genome.

Which scoring matrix was used? BLOSUM45? 62? 80? Without knowing this, the E-value threshold (0.0001) is difficult to ascertain. Other important parameters?

Response 5: We now state explicitly that protein sequences of known drug targets were obtained from DrugBank (approved drug targets). The total number of unique target proteins and their cognate compounds are included in the Methods section entitled 'Cheminformatic pipeline'. We now also provide the full BLASTP command and parameters used for searches against the *A. caninum* proteome (e.g. ancylostoma_caninum.PRJNA72585.WBPS19.protein.fa): blastp (NCBI

BLAST+ 2.10.0+); Scoring matrix: BLOSUM62; E-value cutoff: 1e-4; Low-complexity filtering: SEG enabled (default); Gap penalties: defaults for BLOSUM62. These details are now explicitly stated so that the E-value threshold is interpretable in context on lines 237- 256 of the Methods section as follows:

“A majority vote across the models was used to prioritize approved compounds from the drug database DrugBank (Wishart *et al.*, 2018; Knox *et al.*, 2024). This ensemble approach reduced false positives and improved overall prediction reliability. To improve the specificity of predictions and reduce off-target effects, predicted drug targets were compared against the *A. caninum* genome (PRJNA72585 WBPS19) (Howe *et al.*, 2017). Protein sequences of known drug targets from all the available approved DrugBank compounds were downloaded from DrugBank, and then filtered to only include those that passed the majority vote from the classification algorithms; the target proteins from this filtered list were subjected to BLASTP (Camacho *et al.*, 2009) searches against the *A. caninum* genome. This equated to 2470 unique compounds and 2966 unique targets, and after filtering for compounds that passed the majority vote became 1151 unique compounds. BLAST was performed using local blastp (NCBI BLAST+ 2.10.0+) executable with the following parameters: default scoring matrix: BLOSUM62; E-value cutoff: 1e-4; low-complexity filtering: SEG enabled (default); gap penalties: defaults for BLOSUM62. Matches with e-values > 0.0001 were discarded, retaining only those drug candidates whose targets had significant sequence similarity to hookworm homologs. This lowered the filtered list of compounds from 1151 to 1052. Next, we mapped these 1052 DrugBank compounds to ChEMBL and retained only approved, small compounds that exhibited zero Lipinski violations, which then yielded 478 compounds (Table S4). This dual-layer screening process generated a refined list of high-confidence candidates for phenotypic screening against MADR hookworm”.

Furthermore, the scripts used to extract majority-vote targets, run BLAST against the *A. caninum* database, filter hits by E-value, and map results back to DrugBank IDs and UniProt IDs are available in the BiologyOpen-BLAST repo (<https://github.com/dohalloran/BiologyOpen-BLAST> e.g. `extract_majorityvote_targets.py`, `run_blast_and_filter.sh`, `filter_blast_and_map_drugbank.py`). We also provide the actual database we generated for transparency also within the BiologyOpen-BLAST repo. We reference these scripts and the directory structure in a dedicated README in the repository.

Comment 6. *Lack of data (output, results)... Please provide an annotated notebook or code...*

Please provide plots of ROC curves, precision-recall curves, plots of F1 scores... As for any machine learning paper, these plots and metrics are what the readership expects...

Response 6: As mentioned above we have now included two GitHub repositories:

<https://github.com/dohalloran/BiologyOpen-ML> and

<https://github.com/dohalloran/BiologyOpen-BLAST> and link to them in the revised manuscript under the “**Data and Resource Availability**” section. We also provide details on the classification algorithms performance by including new Figures in the revised manuscript that include performance metrics based on sensitivity (recall), specificity, precision, accuracy, F1 scores, AUC (**Figures S2-S3**), and a multi-panel confusion matrix figure that shows the number of true positives, false positives, true negatives, and false negatives for each classifier (**Figure S4**). We describe these data in the Methods of the revised manuscript on lines 235- 237 as follows: “Model performance was examined based on sensitivity (recall), specificity, precision, accuracy, F1 scores, AUC (Figs S2-S3), and the confusion matrix for each classifier (Fig. S4)”.

Comment 7. *Lack of a simple baseline for comparison... The performance of machine- learning vs simple chemical similarity can be tested and compared... How many TPs, TNs, FPs, FNs are there in each case? Providing a simple baseline for comparison is a nice way to show the improved performance of machine learning methods... How does flutamide rank in each case?*

Response 7: Firstly, we now include some details about flutamide scoring in terms of model prediction. Flutamide yielded the following score for each model (1 or 0 indicates whether it was predicted to be a candidate anthelmintic or not, and this is followed by a probability score): SVC linear: 1, 0.755; KNN: 1, 0.6; SVC RBF: 1, 0.82; NB: 1, 0.99; RF: 0, 0.42; Decision Tree: 0, 0. From these results, we can see that flutamide was predicted to be a possible anthelmintic by four of the six models, thus passing the majority vote in addition to zero Lipinski violations, and thus moving it forward for further analysis. Note: these data are

now all provided in the GitHub repo <https://github.com/dohalloran/BiologyOpen-ML> in the file named 'DrugBank_approved_predictions_all_models.csv'.

Comment 8. *BLAST searches against the A. caninum genome... you may also be interested in trying to capture more distant hits by using profile-HMM searches... This may or may not reveal additional conserved targets but it may be worth trying. This is just a suggestion.*

Response 8: We appreciate this suggestion and agree that employing HMM could be another strategy to identify robust matches to *A. caninum*. We settled on using BLAST for our analyses, but it may be interesting for future work to compare this with HMM profile searches to incorporate more matches. By making all of our BLASTP parameters, inputs, and scripts available in the BiologyOpen-BLAST repo (<https://github.com/dohalloran/BiologyOpen-BLAST>), we have made it very easy for others to adapt/extend this pipeline to detect more distant homologues, potentially revealing additional conserved targets in *A. caninum* and related helminths. We acknowledge this as an attractive future direction, but we keep it as forward-looking rather than adding a full new HMM analysis at this stage.

Comment 9. *Also related to the targets of compounds: you use this as a filter strategy but then do not mention anything about the targets (e.g. which is the target of flutamide? what is the retrieved hit from the A. caninum genome? why do you say that "While its mechanism of action in nematodes remains unknown, its consistent phenotypic effects suggest interference with developmental or neuromuscular pathways across egg-hatching and larval developmental stages."... What are the hypothesized activities for the A. caninum candidate target? Is it conserved in other helminths?*

Response 9: We thank the reviewer for this helpful comment and agree that the targets underlying the phenotypes deserve to be presented more explicitly. Flutamide is a non-steroidal anti-androgen whose primary mammalian target is the androgen receptor (AR, NR3C4: UniProt ID P10275). In our target-based filtering, we used the annotated human AR protein sequence as the query and performed BLASTp against the predicted *A. caninum* proteome. This analysis recovered a candidate nuclear hormone receptor in *A. caninum* (gene ANCCAN_07383 located on contig ANCCANDFT_Contig77:43780-48172). We now explicitly state in the Results section on lines 127-128 and the Discussion section on lines 186-194 that this nuclear hormone receptor is the top-scoring *A. caninum* candidate match for the flutamide target based on our BLAST analysis. We also list all BLAST matches in the BiologyOpen-BLAST repo (<https://github.com/dohalloran/BiologyOpen-BLAST>) in the file named 'MajorityVote_targets_vs_ancylostoma.tsv'.

Although flutamide's target is well-characterized in mammals, there is no direct biochemical or genetic validation in parasitic nematodes confirming that this *A. caninum* nuclear hormone receptor is indeed the functional target. The ortholog of this gene in *C. elegans*, is *nhr-69*, which has been implicated in insulin signaling (PMID: 22359515). Insulin signaling has a well characterized role in growth and development in *C. elegans* (PMID: 24395814). Taken together, we propose a model whereby flutamide could inhibit a regulator of insulin signaling in *A. caninum* which perturbs hatching and development as observed in our data. However, further testing would obviously be required to flesh this out. We now include a discussion of this mechanism on lines 186-212, which includes details on conservation of ANCCAN_07383 with other nematodes, including 67% identity at the protein level with NHR-69 in *C. elegans* and 93% protein identity to the *Haemonchus contortus* (ruminant parasite) ortholog, HCON_00017410.

Comment 10. *Lipinski violations... it would be a nice addition to the paper to analyze and show the features and properties of known anthelmintics in relation to Lipinski rules before applying filters... Perhaps there are some violations to Lipinski's rules that may be admitted or less penalized in the case of anthelmintics?*

Response 10: We thank the reviewer for this suggestion; it aligns well with our interest in positioning anthelmintics within broader drug-likeness frameworks. However, to the best of our knowledge, and based on our analysis, almost all characterized anthelmintics exhibit zero Lipinski violations. The one exception that we are aware of is Moxidectin which has a molecular weight of around 639 Da (above the 500 Da limit). Therefore, based on these observations we felt that removing compounds that exhibit Lipinski violations is a reasonable approach in our pipeline search for repurposed anthelmintics.

REVIEWER 2

Comment 1. Please provide sufficient details on the machine learning methods... number of active vs. decoy sets... preprocessing pipeline, model performance metrics, validation... full list of top predicted compounds... cheminformatics methods should precede the three-tiered phenotypic assay...

Response 1: We appreciate this comment and have substantially expanded and reorganized the Methods and Supplementary Information to address it. In the revised Methods section entitled “Cheminformatic pipeline” (now moved to precede the description of the three-tier phenotypic assay, as suggested), we explicitly describe the construction of the active and inactive (decoy) sets, including the exact numbers of compounds in each group. We reiterate that the active set was curated from known anthelmintic and anti-parasitic compounds, whereas the inactive set was generated by mining PubChem for compounds that were not annotated with “anthelmintic” or “anti-parasitic” and that matched the Tanimoto similarity distribution of the active dataset. All individual compounds in both sets are now listed in **Supplementary Table S1**, and their corresponding descriptor matrices are provided in **Supplementary Tables S2** (actives) and **Table S3** (inactives), in addition to their Tanimoto similarity distributions (**Figure S1**). We further clarify the preprocessing pipeline in detail.

Specifically, we now state the exact descriptor calculation steps using RDKit on lines 224-229. We also reference the fully reproducible descriptor-generation scripts and configuration files in the BiologyOpen-ML GitHub repository, where Section 4 (“Generating RDKit descriptors”) documents the precise commands and parameters used.

In response to the reviewer’s request for more information about the machine-learning methods themselves, we have expanded the Methods to list each classification algorithm and its key parameters. In the revised text (lines 230-237), we now explicitly state that we evaluated a linear Support Vector Classifier, an SVC with RBF kernel, k-Nearest Neighbors ($k = 5$, Minkowski metric, $p = 2$), Gaussian Naive Bayes, a Random Forest classifier ($n_estimators = 10$, $criterion = \text{“entropy”}$, $max_features = \text{“auto”}$), and a Decision Tree classifier ($criterion = \text{“entropy”}$), all implemented in scikit-learn. The training script (`train_activity_models.py`) and helper modules in the BiologyOpen-ML repository contain the exact model definitions, preprocessing steps, and evaluation routines used in the manuscript.

We also now provide a much more complete description of our validation strategy and performance metrics. In the Methods (lines 233-242) we explicitly state that the dataset was split into 80% training and 20% held-out test data ($train_test_split, test_size = 0.2, random_state = 0$) and that 5-fold cross-validation was applied to the training set to estimate performance and tune model selection. In the revised Results and **Supplementary Figures S2-S4**, we report for each classifier the sensitivity (recall), specificity, precision, accuracy, F1 score, and AUC, and we include a multi-panel confusion-matrix figure that shows the number of true positives, false positives, true negatives, and false negatives for each model. The underlying performance metrics and per-compound predictions are provided in the BiologyOpen-ML repository.

Finally, in response to the request for a full list of top predicted compounds, we now supply a complete list of all DrugBank-approved molecules evaluated by our models, together with their per-model predictions and consensus (majority vote) classification. This is provided as a new supplementary table (**Supplementary Table S4**) and as the CSV file “DrugBank_approved_predictions_all_models.csv” in the BiologyOpen-ML GitHub repository. In the Methods section (lines 253-256), we explicitly refer to this table, and we clarify how compounds such as flutamide were prioritized based on majority vote, predicted activity, and Lipinski filter criteria before entering the three-tier phenotypic assay pipeline.

Comment 2. If WMD was maintained by continuous passage from 2013, what is the passage number in 2024... Passaging an organism continuously for over 10 years may not follow best practice...

Response 2: We appreciate the reviewer’s point about long-term passaging. WMD is not a mutagenized laboratory strain but a wild-type canine hookworm strain that must be maintained in its natural host to preserve the life cycle. For obligate parasitic nematodes such as *Ancylostoma*

there is no practical alternative to serial passage through dogs; long-term in vitro culture is not yet a feasible option and cryopreservation produces very mixed results.

WMD has been maintained in our facility by routine passage through purpose-bred dogs since 2013. We do not assign a formal “passage number” to this strain, but at each passage we deliberately infect. We agree that long-term maintenance can, in principle, allow laboratory adaptation. In practice, across this 10-year period, as well as over 35 years maintaining *A. caninum* in lab animals, we have not observed systematic changes in key phenotypes we monitor (e.g. infectivity in dogs, fecundity, life cycle parameters, virulence, and baseline drug sensitivity). We therefore regard WMD as a “laboratory wild-type” reference strain, analogous to long-maintained (decades in some cases) wild-type lines in other nematode systems such as *C. elegans*.

Comment 3. Specify the volume of BU and concentrations for egg hatch assay... details of washes, RPMI volume, antibiotics and antifungal... rationale for using *A. ceylanicum* adults instead of *A. caninum*...

Response 3:

We thank the reviewer for raising this point. We have added the details related to the BU and RPMI in the Methods section on lines 281-282, lines 291-292, and lines 308-311. In terms of the rationale for using *A. ceylanicum*: adults were required for the ex vivo adult drug- response assays, which can only be obtained by terminally collecting worms from the host intestine. For *A. caninum*, this would have required euthanasia of purpose-bred dogs solely to obtain adult worms for these experiments. Our institutional animal care and use committee strongly encourages adherence to the 3Rs (Replacement, Reduction, Refinement), and we therefore chose to use *A. ceylanicum* adults, which can be maintained and collected from experimentally infected rodents rather than dogs. *Ancylostoma ceylanicum* and *A. caninum* are very closely related congeneric hookworm species with highly similar biology, including gut niche, life cycle, and drug targets, and *A. ceylanicum* is widely used as a surrogate experimental model for hookworm infection. We therefore expect the adult-stage responses we measured to be broadly representative of adult *Ancylostoma* biology. We also included a paragraph explaining this rationale in the results section on the ex-vivo adult assays on lines 149-156 as follows: “Next, flutamide was evaluated in ex vivo adult survival assays using the hookworm *A. ceylanicum*. Adult-stage ex vivo assays were performed in *A. ceylanicum* rather than *A. caninum* as they are easily obtained from infected rodents and very closely related to *A. caninum*; using *A. ceylanicum* maintained in rodents allowed us to reduce dog use while still working with a closely related *Ancylostoma* species. Because adult worms are limiting, we focused on the concentrations that showed the strongest effects in the EHA and LDA (50 and 100 μM), alongside 1% DMSO and 10 nM emodepside controls, using one female and two males per condition”.

Comment 4. Gap between protocols and results... focus on flutamide only... need positive-control EHA/LDA curves... raw data and full lists... avoid redundancy in LDA text...

Response 4:

We now include the thiabendazole data in our EHA for the multidrug resistant hookworm strain and include these data on lines 133-135 as follows: “Thiabendazole was also tested on BCR in this trial where we observed zero hatching at 12.5 μM , 25 μM , and 50 μM across three replicates that included 144, 142, and 146 eggs respectively”. However, the main goal of our research is to identify novel anthelmintics that will affect multidrug resistant hookworm, and so using a robust positive control with similar features to flutamide was not possible.

However, as noted in the Methods and Results sections (see Table 1 and Fig. 4) we always include negative controls and now include the thiabendazole data as noted above.

We have also expanded our discussion on flutamide and mechanism of action where we explain that a candidate nuclear hormone receptor is the top-scoring *A. caninum* candidate match for the flutamide target based on our BLAST analysis. We also list all BLAST matches in the BiologyOpen-BLAST repo (<https://github.com/dohalloran/BiologyOpen-BLAST>) in the file named ‘MajorityVote_targets_vs_ancylostoma.tsv’.

All of this is included in the Discussion section on lines 186-197 as follows: “From our BLAST analysis using the annotated human androgen receptor protein sequence as a query against the

predicted *A. caninum* proteome, we identified a candidate nuclear hormone receptor in *A. caninum*, gene ANCCAN_07383, as the top hit. ANCCAN_07383 exhibits 67% identity at the protein level with NHR-69 in *C. elegans* and 93% protein identity to the *Haemonchus contortus* (ruminant parasite) ortholog, HCON_00017410. Although flutamide's target is well-characterized in mammals, there is no direct biochemical or genetic validation in parasitic nematodes confirming that this *A. caninum* nuclear hormone receptor is indeed the functional target, however, the ANCCAN_07383 ortholog in *C. elegans*, *nhr-69*, has been implicated in insulin signaling (Park *et al.*, 2012). Insulin signaling has a well characterized role in growth and development in nematodes (Brand and Hawdon, 2004; Murphy and Hu, 2013). Taken together, we propose a model whereby flutamide could inhibit a regulator of insulin signaling in *A. caninum* which perturbs hatching and development as observed in our data”.

In the revised manuscript we have revised the results section on the LDA assay to reduce any redundancy, which is now on lines 137-147 as follows: “To determine whether flutamide affects later stages of development, LDAs were performed using the drug-resistant BCR isolate. Flutamide inhibited L1-L3 development with an IC₅₀ of 42.3 μM (Table 1). In controls, 84.8% of L1 developed to L3 after 7 days, with 9.2% at L2 and 6.0% at L1. Low flutamide concentrations (2-20 μM) produced a similar distribution (≥86% L3; ≤14% L1-L2). In contrast, at 50 μM no larvae reached L3, with 83.7% arrested at L2 and 16.3% at L1, and at 100 μM 19.4% remained at L2 and 80.6% at L1. Shapiro-Wilk tests indicated that percentage data for each stage deviated from normality (L1 W=0.654, *p*=0.0029; L2 W=0.668, *p*=0.0044; L3 W=0.697, *p*=0.0089), so Dunn's test was used for multiple comparisons. Higher flutamide concentrations (50 and 100 μM) significantly altered the distribution of larvae across stages compared with 2 and 20 μM (*p* < 0.05 for all stages), demonstrating a potent inhibitory effect on development to L3 at 50-100 μM and to L2 at 100 μM”.

Finally, we also focus our results specifically on flutamide and have revised the initial screening section using the egg hatch assays (EHA) to reflect this on lines 125- 133 as follows: “Phenotypic screening began with EHA using eggs isolated from dogs infected with the BCR isolate of *A. caninum* (Fig. 3). Flutamide emerged as a lead candidate, significantly inhibiting egg hatching in a concentration-dependent manner (Fig. 4). Flutamide is a non-steroidal anti-androgen whose primary mammalian target is the androgen receptor (UniProt ID P10275). The calculated IC₅₀ was 65.62 μM for drug resistant hookworm, with near-complete inhibition observed at 100 μM (Fig. 4, yellow line). No visible hatching or embryonation was detected at this dose (100 μM). Three biological replicates confirmed this result, and the compound demonstrated low variability between trials. Wildtype WMD *A. caninum* eggs were also included and yielded an IC₅₀ of 129.7 μM (Fig. 4, blue line)”.

Comment 5. *Improve readability and interpretation of Figures 1 and 2 (legends, colour scales, axes, cluster names/structures)... label flutamide IC50 values visually in Figure 4... present adult worm survival data.*

Response 5: We appreciate this comment and have substantially revised Figure 1 and the legend to make the pipeline easier to follow and interpret. In the original submission, the legend for Figure 1 compressed multiple computational and filtering steps into a relatively dense diagram, which may have obscured how the machine-learning and genomic components fit together. In the revised Figure 1 and legend, we now present the pipeline as a clearly ordered sequence of steps that mirrors the structure of the Methods section: (i) curation of active and inactive compounds from PubChem; (ii) calculation of RDKit molecular descriptors and assembly of feature matrices; (iii) training and evaluation of multiple scikit-learn classifiers (including SVMs, k-nearest neighbors, Naive Bayes, random forest, and decision tree); (iv) majority-vote prioritization of DrugBank-approved compounds; (v) BLASTp-based target homology filtering against the *A. caninum* proteome; and (vi) mapping and filtering of the resulting DrugBank compounds to obtain the final set of candidates for experimental screening. We believe these changes significantly improve the readability and interpretability of Figure 1 and more clearly convey how the different computational components contribute to the final candidate list.

In relation to Figure 2, we also agree that the visualization of the compound space and the position of flutamide could be made clearer. In the revised manuscript, we have added a new

Figure 2 plot that provides two complementary visualizations based on the underlying SDF data. In Figure 2A, we now show a pairwise distance heatmap in which distances are computed directly from atom-count vectors (Euclidean distance on atom counts). Compounds are ordered by hierarchical clustering, and the color scale is explicitly defined (dark blue = more similar; dark red = more dissimilar) to aid interpretation. Importantly, flutamide (PubChem CID 3397) is highlighted directly on the heatmap by a thickened black line, allowing the reader to immediately locate flutamide and visually assess its distance to every other compound. In Figure 2B, we show a three-dimensional atom-pair MDS representation derived from atom-pair fingerprints. Flutamide is emphasized as the only labeled point in this space, plotted as a larger triangle with an arrow and text label. Together, these changes substantially improve the readability of Figure 2 and make the chemical relationships between flutamide and the rest of the library more transparent.

Finally, we also added explicit IC_{50} labels for flutamide in WMD and BCR assays to Figure 4 as well as the negative control line, and now provide more discussion presenting the adult worm survival data and raw counts in the Results section on lines 149- 161 as follows: “Next, flutamide was evaluated in *ex vivo* adult survival assays using the hookworm *A. ceylanicum*. Adult-stage *ex vivo* assays were performed in *A. ceylanicum* rather than *A. caninum* because adult *A. caninum* can only be obtained by euthanizing infected dogs; using *A. ceylanicum* maintained in hamsters or mice allowed us to reduce dog use while still working with a closely related *Ancylostoma* species. Because adult worms are limiting, we focused on the concentrations that showed the strongest effects in the EHA and LDA (50 and 100 μ M), alongside 1% DMSO and 10 nM emodepside controls, using one female and two males per condition. At both 24 and 48 h, all three worms in the DMSO control were alive. In the emodepside group, all three worms were dead at each time point. At 50 μ M and 100 μ M flutamide, all three worms were alive at 24 h and at 48 h. Thus, emodepside reduced viability from 100% to 0% (3/3 worms dead) at both 24 and 48 h, whereas flutamide at 50 or 100 μ M did not significantly reduce adult viability relative to the DMSO control (Fisher’s exact tests: two-sided $p > 0.05$; one-sided $p > 0.05$)”.

Comment 6. *Include possible reasons for flutamide activity... compare IC_{50} values vs standard anthelmintics... discuss physiological relevance of up to 100 μ M... explain extrapolation from *A. ceylanicum* to *A. caninum*, or acknowledge limitation... propose future research directions.*

Response 6: We thank the reviewer for these insightful suggestions, which have prompted us to expand both our mechanistic discussion of flutamide and our interpretation of its potency and translational relevance. In the revised Discussion (lines 186-212), we now explicitly connect flutamide’s observed activity in our assays to its known pharmacology and to our target-mapping analysis. As mentioned above in our response to Reviewer 1 Comment 9, but reiterated here for clarity, flutamide is a non-steroidal anti-androgen whose primary mammalian target is the androgen receptor (AR/NR3C4; UniProt P10275). Using this human AR sequence as a query in our BLASTP-based pipeline, we identified a candidate nuclear hormone receptor in *Ancylostoma caninum* (gene ANCCAN_07383) as the top-scoring putative ortholog. We now clearly state that ANCCAN_07383 is the *A. caninum* candidate target recovered by our BLAST analysis and that this protein shows substantial conservation with nuclear hormone receptors in other nematodes, including 67% identity with the *Caenorhabditis elegans* ortholog NHR-69 and 93% identity with the *Haemonchus contortus* ortholog HCON_00017410. We have added a paragraph proposing a mechanistic model in which flutamide perturbs a nuclear hormone receptor involved in insulin signaling and developmental regulation, drawing on prior work implicating NHR-69 in insulin pathways and the well-established role of insulin signaling in nematode growth and development. We emphasize that this model is currently hypothetical and that there is, as yet, no direct biochemical or genetic evidence that ANCCAN_07383 is the functional flutamide target in *A. caninum*, and we explicitly flag this as a key avenue for future investigation.

To address the reviewer’s request for a more quantitative contextualization of flutamide’s activity in relation to its IC_{50} values and the concentrations used: we agree that it is important to place flutamide’s activity in the context of established anthelmintics and to comment on the relevance of the relatively high concentrations used in our *in vitro* assays. In our assays flutamide acts in the tens-of-micromolar range. By contrast, both in our hands and in the published literature, standard anthelmintics are active at much lower concentrations. For example, our positive-control drug emodepside is fully active at 10nM in the adult *A. ceylanicum* survival assay,

and macrocyclic lactones such as ivermectin, as well as benzimidazoles, typically show IC₅₀/EC₅₀ values in the low-micromolar to nanomolar range in nematode motility or development assays. Thus, flutamide is at least one to several orders of magnitude less potent than currently used anthelmintics in comparable *in vitro* systems. Regarding physiological relevance, human pharmacokinetic data for flutamide show that after standard oral doses (250-500 mg), peak plasma concentrations of the parent drug are only ~0.02-0.1 µg/mL (≈0.07-0.36 µM), while the active metabolite 2-hydroxyflutamide reaches ~1.3-2.4 µg/mL (≈4-8 µM), with steady-state levels around 0.94 µg/mL (~3 µM) (PMID: 3169114). These exposures are around an order of magnitude lower than the IC₅₀ values we observe *in vitro*, and far below our highest test concentration of 100 µM. Although local drug levels in the intestinal lumen after an oral dose can transiently exceed plasma concentrations, we agree that 100 µM is above what is likely achievable systemically in humans with currently approved flutamide dosing. However, flutamide could well represent a chemical probe that reveals a druggable pathway controlling hookworm larval development. Higher-potency analogues or different chemotypes targeting the same pathway would be required for translational development. We have added text related to this in the Discussion on lines 205-212 as follows: “After standard oral doses in humans, flutamide and its active metabolite 2-hydroxyflutamide reach peak plasma concentrations in the low micromolar range (≈0.1 µM and 4-8 µM, respectively) (Schulz *et al.*, 1988), which is well below the IC₅₀ values observed in our assays. While intestinal luminal concentrations may transiently be higher than plasma, we do not regard 100 µM as a clinically realistic exposure. Our data therefore indicate that flutamide serves primarily as a tool compound that reveals a hormonally regulated developmental pathway in hookworms, rather than as a direct candidate for therapeutic repurposing”.

Regarding extrapolation from *Ancylostoma ceylanicum* to *A. caninum*, please see our response to Comment 3 above.

MINOR COMMENTS:

Comment: *Italicize all the in vivo and in vitro where they appeared in the study.*

Response: All instances of *in vivo* and *in vitro* have been checked and are now consistently italicized.

Comment: *Add the ethical approval number in the ethical standards section.*

Response: We have added the full ethical approval number in the “Ethical standards” section.

Comment: *Justify why compounds not available in solution were excluded, since DMSO was used to solubilize the compounds.*

Response: We agree that DMSO is a useful general solubilizing vehicle and it was used for all compounds that we could bring into solution under our assay conditions. However, several candidate compounds could not be obtained at sufficient purity or quantity, or remained insoluble/precipitated even in DMSO at concentrations compatible with our assays (i.e. within the maximum DMSO percentage that does not affect worm viability). For some compounds, achieving the desired concentrations would have required higher DMSO levels or additional co-solvents that were incompatible with the biological readout. We therefore restricted our screening to compounds that could be reliably solubilized and maintained in solution under the same assay conditions as the rest of the panel.

Comment: *Use Microsoft Equation Editor to represent the formula for calculating the percentage of developed larva ($L3_{total} / (L1 + L2 + L3) \times 100$).*

Response: The formula is now displayed in equation format in the Methods.

Second decision letter

MS ID#: bio.062380R1

MS Title: Drug repurposing to combat multidrug-resistant hookworm

Authors: Elise L. McKean, Catherine A. Jackson, Abigail Dowd, John M. Hawdon and Damien O'Halloran

I am happy to tell you that your manuscript has been accepted for publication in Biology Open, pending our standard publication integrity checks. It was accepted on 16th January 2026.